# Sulodexide Develops Contraction in Human Saphenous Vein via Endothelium-Dependent Nitric Oxide Pathway

**DOI:** 10.3390/jcm12031019

**Published:** 2023-01-28

**Authors:** Suat Doganci, Mehmet Emin Ince, Meric Demeli, Nadide Ors Yildirim, Bilge Pehlivanoglu, Alperen Kutay Yildirim, Sergio Gianesini, Yung-Wei Chi, Vedat Yildirim

**Affiliations:** 1Department of Cardiovascular Surgery, Gulhane Training and Research Hospital, University of Health Sciences, Ankara 06010, Turkey; 2Department of Anesthesiology and Reanimation, Gulhane Training and Research Hospital, University of Health Sciences, Ankara 06010, Turkey; 3Department of Physiology, Hacettepe University Faculty of Medicine, Ankara 06100, Turkey; 4Department of Cardiovascular Surgery, Gazi University Faculty of Medicine, Ankara 06560, Turkey; 5Translational Medicine Department, University of Ferrara, 44121 Ferrara, Italy; 6Department of Surgery, Uniformed Services University of the Health Sciences, Bethesda, MD 20814, USA; 7Vascular Center, University of California, Sacramento, CA 95817, USA

**Keywords:** sulodexide, L-NAME, Krebs–Henseleit, saphenous, vein, contraction, venous disease

## Abstract

Chronic venous disease (CVD) is a proqgressive and underestimated condition related to a vicious circle established by venous reflux and endothelial inflammation, leading to vein dilation and histology distortion, including loss of media tone. Sulodexide (SDX) is a drug restoring the glycocalyx that demonstrated endothelial protection and permeability regulation, together with anti-thrombotic and anti-inflammatory roles. In the lab it also exhibited vein contractility function. The aim of the present study was to show the possible role of endothelium and nitric oxide pathway on SDX’s veno-contractile effect on human saphenous veins. The remnants of great saphenous vein (GSV) segments (*n* = 14) were harvested during coronary artery bypass graft surgery. They were dissected as endothelium-intact (*n* = 8) and denuded rings (*n* = 6). First, a viability test was carried out in bath with Krebs–Henseleit solution to investigate a control and basal tension value. After this, cumulative doses of SDX were applied to rings and contraction values were studied in endothelium-intact phenylephrine (PheE, 6 × 10^−7^ M) pre-contracted vein rings. Finally, endothelium-intact PheE pre-contacted vein rings were treated by nitric oxide synthase inhibitor Nω-nitro-L-arginine methyl ester (L-NAME, 10^−4^ M) for 10 min. Contraction protocol was applied, and contraction values were measured in cumulative doses of SDX. The same protocol was applied to endothelium-denuded vein rings to investigate the effect of SDX. Saphenous vein rings showed an increase in contraction to cumulative doses of SDX. In endothel-intact rings, KCL-induced contraction from 92.6% ± 0.3 to 112.9% ± 0.4 with cumulative SDX doses. However, SDX did not show any veno-contractile effect on endothel-denuded rings. In denuded rings contraction responses measured from 94.9% ± 0.3 to 85.2% ± 0.3 with increasing doses of SDX, indicating no significant change. Nitric oxide synthase inhibitor (L-NAME) prohibited the contraction response of the sulodexide in all dosages, indicating that the contractile function of SDX was mediated by endothelial derived nitric oxide. Results of endothel-intact and denuded rings with L-NAME showed a similar incline with denuded rings with SDX only. The results confirmed SDX’s veno-contractile effect in human samples, by means of nitric oxide synthase pathways involvement.

## 1. Introduction

Sulodexide (SDX) is a purified combination of glycosaminoglycans obtained from porcine intestinal mucosa [1]. Glycosaminoglycans contain either sulfated or non-sulfated monosaccharides. They have many important roles, such as the regulation of protein activities via affecting cytokines, adhesion molecules, and chemokines, in addition to antiproteolytic effects [2].

SDX has repeated disaccharide units, forming a molecule containing unbranched polysaccharide chains [3]. It consists of 80% heparan sulfate and 20% dermatan sulfate [4]. Since the effect of heparan sulfate on coagulation is minor compared to unfractionated heparin (UFH) or low molecular-weight heparin (LMWH), the risk of hemorrhage is lower [5]. Its high affinity for antithrombin and longer half-life is associated with oral bioavailability [4,5]. The dermatan sulfate component is made up of iduronic acid and galactosamine, mainly found in the vascular walls and endothelium [6]. It exhibits anticoagulant activity inhibiting factor X and II [7,8]. This immediate anticoagulant action of SDX is less potent than UFH and LMWH, however, its unique anticoagulant effect becomes prominent over time [8]. Pharmacokinetic studies of SDX demonstrated that the distribution volume of the drug is very large due to its affinity for the surface of endothelium rather than for the plasma proteins [9].

SDX as a glycosaminoglycan protects the endothelial glycocalyx, which coats the vascular endothelial lumen. The glycocalyx layer also provides an antithrombotic and profibrinolytic surface in addition to its role in management of vascular tone, permeability, and response to shear stress. Heparan sulfate protects both the endothelium and glycocalyx from reactive oxygen species and downregulates cytokines. Moreover, it reconstructs the glycocalyx by binding directly to endothelium [3,10,11].

Summing up, SDX’s antiplatelet, anti-inflammatory, antiproteolytic, connective, and endothelial tissue protective effects have been demonstrated in several studies [12,13,14,15,16,17,18,19,20,21,22], leading to therapeutic indication in both arterial and vascular disease management [23].

Chronic venous disease is a progressive and underestimated condition. It has huge socioeconomical, psychological, and physical impacts on the population. The estimated incidence of chronic venous disease scales up to 80% [24]. Its prevalence is higher in women [24]. The broad spectrum of clinical manifestations can vary from asymptomatic venous hypertension and varicose veins to edema, skin changes, and leg ulcers, and it is associated with complex pathophysiological mechanisms. Chronic venous disease refers to a broad spectrum of abnormalities and is associated with complex pathophysiological mechanisms. The interaction between environmental and genetic backgrounds is responsible for increased ambulatory venous pressure, which eventually alters the structure and function of the venous system [25]. The pathophysiological mechanism of chronic venous disease mainly involves an increase in ambulatory venous pressure and dilatation of veins due to continuous reflux from incompetent valves and venous obstruction [26]. The hemodynamic impairment leads to endothelial inflammation by biomechanical pathological transduction, generating a proteolytic environment and a vicious cycle of progressive endothelial and hemodynamic deterioration [27]. This inflammatory cascade creates an infiltration of macrophages, cytokines, and an increase in level of metalloproteinases (MMPs) [28]. The mechanical stress caused by increased reflux also deteriorates the endothelial glycocalyx. It increases production of reactive radicals, further increasing the inflammation. A turbulent flow with increased venous pressure generates the leukocyte adhesion and proteolytic enzyme release. Through this cascade, there is also a release of inflammatory mediators such as MMPs, chemokines, cytokines, vascular cell adhesion molecule (VCAM-1), and vascular endothelial growth factor (VEGF). Once these pathophysiological cascades start to flow, the remodeling of vein walls and valves is inevitable [29].

The clinical benefits of SDX have been shown in many studies. Cospite et al. assessed changes in microcirculation by measuring the capillary filtration coefficient [30]. In this study it demonstrated that the group receiving SDX had a lower coefficient, indicating a lower capillary permeability. SDX also has an evident role on venous ulcerations. In addition to its antithrombotic and anti-inflammatory effects, it upregulates the expression of fibroblast growth factors, promoting vascular repair [31]. In the sulodexide arterial venous Italian study (SUAVIS), higher ulceration healing rates were achieved in the SDX group. The time required for healing was shorter compared to the placebo group. Although there is no trial about the role of SDX for the prevention of chronic venous disease, Luzzi et al. evaluated the prevalence of post-thrombotic syndrome after conclusion of anticoagulation for deep-vein thrombosis [32]. According to that study, prevalence of post-thrombotic syndrome in the group that took SDX was lower than the group with standard medical management. Two interesting studies about sulodexide were conducted in animal models aiming to evaluate the response of sulodexide on rat aorta, mesenteric arteries, and inferior vena cava. The studies indicated that SDX promoted arterial relaxation by endothelium-dependent nitric oxide (NO) production, and induced improvement of vein function by causing a decrease in MMP-2 and MMP-9 [12,13]. Its effects on vascular tone were mainly investigated in animal studies using the aorta or inferior vena cava. Since no research has been conducted on human lower-limb veins, which are the main target of treatment in the primary indication of sulodexide, we aimed to investigate the effect of SDX on venous tone, contractile response of the saphenous veins, and the role of endothelial layer. We built a hypothesis that SDX improves the contraction of venous rings via an endothelial-mediated pathway.

## 2. Materials and Methods

The remnants of great saphenous vein (GSV) segments (*n* = 14), harvested during coronary artery bypass graft surgery (CABG), were used in this study. The protocol, approved by the Ethics Committee of Health Sciences University, Gulhane Faculty of Medicine (Issue # 2021-378), was explained to the patients who underwent elective CABG and their informed consent was obtained. The study was conducted in line with the ethical standards of Declaration of Helsinki.

The healthy GSV segments were randomly allocated into endothelium-intact (*n* = 8) or endothelium-denuded (*n* = 6) groups and processed accordingly. The mean age of the patients was 62.4 ± 8.5 in endothelium-intact group and 61.2 ± 9.6 in endothelium-denuded group. The healthy human GSV segments harvested by routine surgical technique (*n* = 14) and the pieces not to be employed in CABG were separated by the surgeon after careful inspection. These pieces were transported to the laboratory in Krebs–Henseleit solution (NaCl, 118; KCl, 4.7; CaCl_2_, 2.5; KH_2_PO_4_, 1.2; MgSO_4_, 1.2 glucose, 10; and NaHCO_3_, 25 mM; pH: 7.4) at 4 °C. Venous segments were dissected gently from the adhering tissue and 3–4-mm-long rings with intact endothelium were prepared. At least four rings were prepared from the GSV segment of each patient. The rings were mounted in a double layered water bath filled with Krebs–Henseleit solution, gassed with 95% O_2_ and 5% CO_2_ at 37 °C. Isometric tension changes were acquired in real time and analyzed using force-displacement transducers (MAY FDT 05, Commat, Ankara, Turkey) and a multichannel computerized data acquisition/analysis system (BIOPAC MP30, Biopac Systems Inc., Goleta, CA, USA). Rings were allowed to equilibrate under a final resting tension of 0.5–1 g for at least one hour, with washouts every 15 min. All the medicine used by the patient was cleared and all the rings became similar during the stabilization period, together with complete washouts in the period from harvesting to mounting; the duration was at least 90 min and the bathing solution was refreshed eight times. After equilibration, the venous rings were challenged with 120 mM KCl to test their viability and to obtain a reference value (KCL MAX). The rings that did not respond were excluded from the study. Following equilibration and regaining resting tension in the first protocol, an escalating dose response curve of SDX (0.001 mg/mL, 0.005 mg/mL, 0.01 mg/mL, 0.05 mg/mL, 0.1 mg/mL, 0.5 mg/mL, and 1 mg/mL) was recorded from the saphenous vein rings (*n* = 8) which were pre-contracted with the α-adrenergic receptor agonist phenylephrine (PheE, 6 × 10^−7^ M). The consecutive SDX doses were applied in five-minute intervals. In the second protocol, another venous ring set (*n* = 8) were pretreated from these patients with nitric oxide synthase inhibitor Nω-nitro-L-arginine methyl ester (L-NAME, 10^−4^ M) for 10 min before stimulation with PheE and further contraction protocol with SDX was applied. The same two data sets were repeated using segments (*n* = 6/group) whose endothelial layers were mechanically scrubbed (endothelium denuded), to investigate the effect of the endothelial layer on response to SDX. One strip was always spared as time control (TC) in each protocol, which underwent KCl and PheE stimulation only and was not exposed to SDX, instead an equal volume (100 µL) of Krebs’ solution was added to the bath fluid to equalize the total volume at the time points of SDX application.

Sulodexide was produced by Alfasigma. The company did not support the investigation by products or funding.

### Calculations and Statistical Analysis

Data were analyzed by using a statistics software package for Windows (SPSS 22.0) (SPSS Inc., Chicago, IL, USA). All values were reported as mean ± standard error of the mean (SEM). The force of contractions was normalized to the wet tissue weight (g/100 mg wet tissue weight), and all the contraction or relaxation responses were presented as the percentages of the KCl-induced maximal contraction. Emax (the highest contraction response induced by SDX) values were calculated by using the SigmaPlot 10.0 for Windows (Systat Software, Inc., San Jose, CA, USA) software.

We first tested the data for normal distribution by the Kolmogorov–Smirnov test. The categorical variables (comorbidities and smoking status) were evaluated by Chi-squared (χ^2^) test. The endothelium-intact and denuded groups were compared by Student’s t test for independent samples. Within comparisons of cumulative concentration-response curves were performed by repeated measures analysis of variance (ANOVA) followed by Tukey’s post-hoc test for multiple comparisons. The differences were considered statistically significant when *p* < 0.05 (two-tailed).

## 3. Results

Demographic data and comorbidities of patients included in the study were given in Table 1. The patients were compared for age, morbidity, and smoking status between endothel-intact and denuded groups.

The groups were comparable in their response to phenylephrine stimulation, however the effect of SDX was different between groups (Figure 1). In vitro SDX application resulted in a concentration-dependent increase in the force of contraction in endothelium-intact saphenous vein rings (*p* < 0.05). However, the force of contraction did not change in SDX-treated endothelium-denuded rings (Figure 1).

The significant venocontractile effect of SDX in endothelium-intact rings was evident at higher doses from 0.01 mg/mL. This difference was valid both for KCl MAX and PheE-induced contractions, furthermore, SDX exhibited a cumulative response resulting in stronger contractions with increasing doses (Table 2). On the contrary, in PheE precontracted endothelium-denuded saphenous vein rings, there was no significant difference in the force of contraction with SDX application (Table 2).

The results of L-NAME (a nitric oxide synthase inhibitor) incubation of the rings prohibited SDX contraction response in all dosages (Figure 2). We obtained vascular tonus similar to that contracted by PheE in all SDX doses. In endothelium-intact rings, pretreatment with L-NAME for 10 min resulted in abolished SDX-induced contraction response (Figure 2, Table 2).

The maximum contraction response obtained from PheE precontracted endothelium-intact and denuded rings were 90.5% ± 0.3 and 95.0% ± 0.3, respectively. The PheE contraction in L-NAME incubated endothelium-intact and denuded rings were 92.3% ± 0.2 and 94.2% ± 0.3, respectively. There was no significant change between endothelium-intact and denuded rings in their response to PheE in both conditions (*p* > 0.05). (Figure 3) The Emax values for SDX were significantly different between endothelium-intact rings (112.9% ± 0.4) and endothelium-denuded rings (94.9% ± 0.3) (*p* < 0.05). On the other hand, the Emax values for SDX in L-NAME incubated rings were comparable (92.5% ± 0.3 and 93.6% ± 0.3 in endothelium-intact and denuded rings, respectively) (*p* > 0.05). (Figure 3).

In line with differences in E max values, the maximum responses were obtained with 1 mg/mL SDX dose in endothelium-intact rings and with 0.001 mg/mL SDX dose in endothelium-denuded rings (*p* < 0.05). The SDX doses where the maximum force of contraction was obtained in L-NAME incubated endothelium-intact and denuded rings were 0.01 mg/mL and 0.001 mg/mL SDX, respectively. These values were not statistically different, however, they were both low compared to the results acquired from endothelium-intact control rings (*p* < 0.05) (Figure 3).

## 4. Discussion

To our knowledge, this is the first study demonstrating the dose-dependent venocontractile action of SDX in human veins, together with its dose dependent effect and endothelial-mediated mechanism of action. The herein presented data represent a significant step forward in the research line involving rat aortas, the mesenteric artery, and inferior vena cava, as animal models [12,13].

The endothelium’s primary role in the SDX’s mechanism of action is of great interest for fostering further comprehension of the pathophysiology of venous disease. It releases many vasoactive factors, such as NO, prostacyclin (PGI2), and the endothelium-derived hyperpolarizing factor (EDHF). NO mediates a cGMP-associated decrease in intracellular Ca^+2^ and reduced the Ca^+2^ sensitivity of contractile proteins, promoting relaxation in vascular smooth muscles. The stimulation of the prostanoid receptor by PGI2 and hyperpolarization of vascular smooth muscle caused by EDHF are other mechanisms for relaxation [33]. A recent study from Raffetto et al. revealed that SDX-induced arterial relaxation was dependent on NO [13]. In this study, L-NAME significantly inhibited the relaxation caused by SDX in rat mesenteric artery and aorta. In another study from Raffetto et al. acetylcholine (Ach) and L-NAME were used with the same contraction protocol to investigate the NO-cGMP pathway on venous relaxation [34]. Ach-induced venous relaxation was significantly inhibited in rings pretreated with L-NAME. According to these findings, venous relaxation was mostly caused by the NO-cGMP pathway. The herein presented study is innovative as it evaluated SDX effect on different vein samples, with and without the endothelium component. According to our study, the venocontractile effect of SDX involved the endothelial-derived nitric oxide pathway. The contractile property was dose-dependent in endothel-intact vein rings (Figure 1). However, in endothelium-denuded rings and rings incubated with L-NAME, there was no veno-contraction effect, and the amount of contraction responses was similar between the two groups (Figure 1). According to findings, it is postulated that SDX required an intact endothelium and nitric oxide synthase mechanism to show a veno-contractile effect. These findings may have occurred as a result of interference to the NO pathway or endothelial vasoactive modulation.

In addition to the NO-cGMP pathway, other mechanisms may also be involved in contraction. Recent research from Raffetto et al. demonstrated that the venous tissue activity of MMPs was intensified in veins under prolonged stretching [35,36]. They also reported the inhibitory effect of MMP-2 and MMP-9 on veinous contraction in a concentration-dependent manner. Loss of tonicity and excessive venodilatation are associated with chronic venous disease. The venotonic effect is a necessity to decrease ambulatory venous pressure and to prevent other pathophysiological changes. The study of Raffetto et al. [37] involved a 2 g-stretch; pretreatment of vein segments with SDX caused a significant decrease in levels of MMP-2 and MMP-9 compared to those without SDX. In another study, the management of chronic venous disease patients by SDX caused a significant decrease in the serum concentration of MMP-9 and MMP-2. SDX also has an inhibitory effect on PheE-induced vein contraction over membrane polarization and K+ channels activity [38]. The reduction of venous contractility was partially reversed when a pretreatment of tissue inhibitor, metalloproteases, was applied [39]. In our study, a resting tension of 0.5–1 g was applied to each ring to mimic the normal average tension that the lower limbs experience, as shown by interval Raffetto et al. to be the best tension for contractile response to be elicited [13]. Considering the significant increase in venous contraction in groups with cumulative doses of SDX compared to PheE only, it can be stated that SDX improves the contractile function of vascular smooth muscles of veins, which makes SDX a good choice for the treatment of patients with venous diseases. The loss of the contraction ability in all doses of SDX whenever the vein endothelium was removed and a nitric oxide synthase inhibitor was applied indicates that a healthy and functional endothelium, specifically the NO pathway, is crucial for the effect of SDX on veno-contraction. In addition to the crucial effect of endothelium and NO pathway that we observed, decreased levels and activities of MMPs may also represent another mechanism responsible for SDX’s venotonic effect. Sulodexide, with its various effects and HS and DS components, is beneficial for peripheral venous insufficiency; however, in the light of our findings, it should be kept in mind that its use may be of limited therapeutic outcome in severe venous diseases where the integrity and/or function of the endothelium is disturbed.

Even though our study contributes significant data to the literature, since it is the first study conducted on human tissue, there were also some limitations. We demonstrated and showed SDX’s venocontractile effect in cumulative doses with an ex vivo experimental design. In order to investigate SDX’s contractile effect on veins more precisely, an in vivo design and clinical studies are necessary. Moreover, we only used healthy saphenous vein specimens in our study. Vein donor patients were not affected by venous hypertension in our study. Studying the involvement of veins in those with CVD in addition to healthy samples may provide more information on SDX’s effect. More comprehensive methods for investigating the pathways may also be used. We only used L-NAME to investigate the NO pathway. 4-amino-5-methylamino-2′,7′-difluorofluorescein diacetate (DAF-FM) is a reagent that is used to detect and quantify low concentrations of NO. It is essentially nonfluorescent until it reacts with NO. DAF-FM, or an analytical chemistry test, Griess, may be used to acquire more detailed information about the NO pathway, especially in the setting of SDX. 

## 5. Conclusions

SDX promotes a concentration-dependent venous contraction in human saphenous veins exposed to laboratory-induced hypertension. The mechanism of this contraction requires an intact endothelium and NO-mediated pathway. Our result demonstrated SDX’s venocontractile effect, in addition to previously known functions such as anti-inflammatory, anti-oedema, and anti-thrombotic properties.

## Figures and Tables

**Figure 1 jcm-12-01019-f001:**
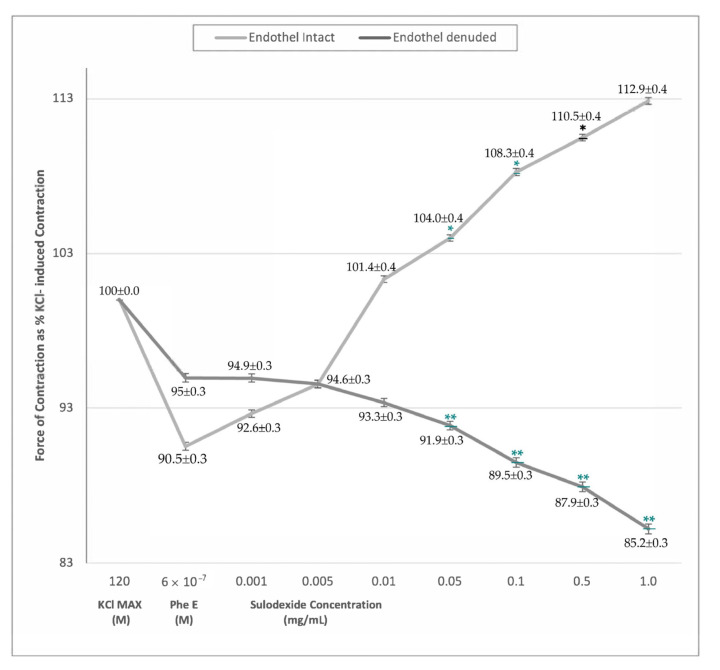
Sulodexide dose-response (0.005 mg/mL^−1^ mg/mL) curves of phenylephrine pre-contracted endothelium-intact and denuded venous rings. Data presented as the percentage of maximum contraction (KCl MAX) induced by KCl (120 mM). * *p* < 0.05 within-group comparison ** *p* < 0.05 between-groups comparison.

**Figure 2 jcm-12-01019-f002:**
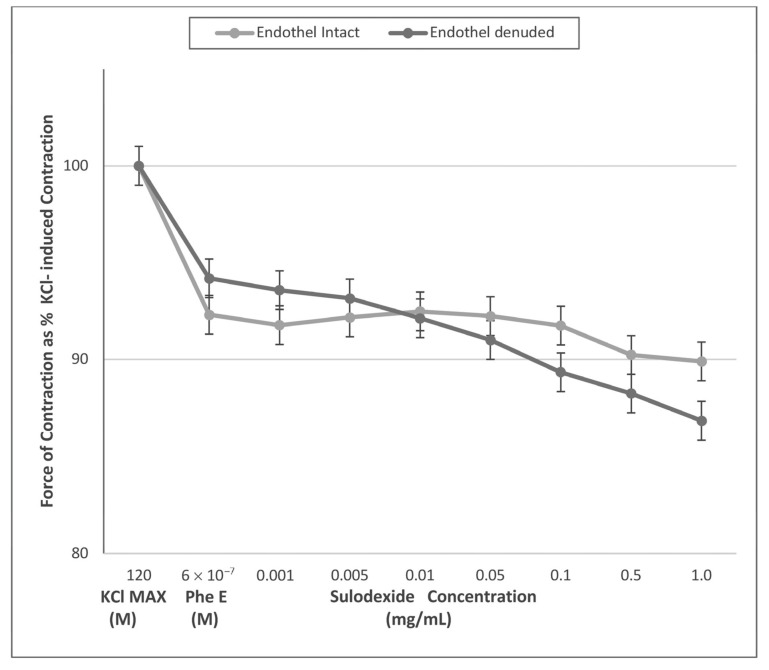
Sulodexide dose-response (0.005 mg/mL^−1^ mg/mL) curves of phenylephrine (PheE) pre-contracted endothelium-intact and -denuded venous rings after L NAME (10^−4^ M) pre-incubation. Data presented as the percentage of maximum contraction (KCl MAX) induced by KCl (120 mM). After incubation with L-NAME, both endothelium-intact and denuded rings showed no changes in terms of contraction to cumulatively increasing doses of SDX (*p* > 0.05).

**Figure 3 jcm-12-01019-f003:**
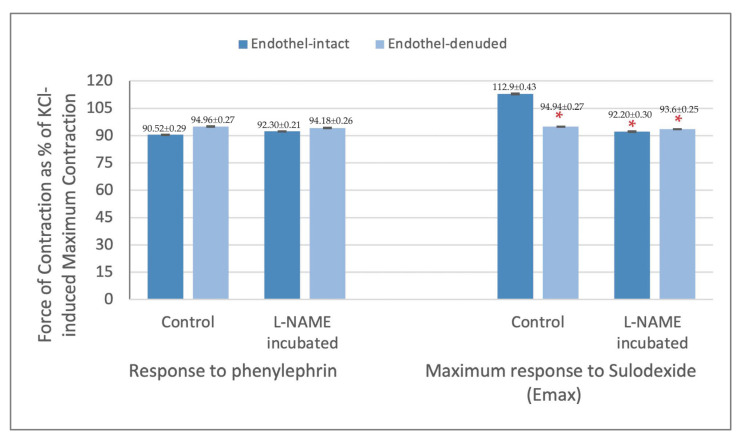
Contraction response of endothelium-intact and denuded saphenous vein strips to phenylephrine (6 × 10^−7^ M) and maximum responses (Emax) obtained with sulodexide application presented as % of KCl-induced contraction. * *p* < 0.05 when compared with Emax obtained from endothelium-intact control rings.

**Table 1 jcm-12-01019-t001:** Demographic data and comorbidities of patients. Age, sex, comorbidities, and smoking habit of the patients whose vein segments were tested. Patients divided into two groups according to endothelium integrity. (DM: diabetes, HT: hypertension, HL: hyperlipidemia, BPH: benign prostate hyperplasia).

Endothel	Patient	Sex	Age	Comorbidities	Smoke
Intact	Patient-1	Male	71	DM, HT, HL	+
	Patient-2	Female	52	DM	−
	Patient-3	Male	68	HT	−
	Patient-4	Male	71	HT	+
	Patient-5	Male	57	NONE	+
	Patient-6	Male	68	DM, HT	+
	Patient-7	Male	50	HT, HL	+
	Patient-8	Male	62	DM, HT	+
Denuded	Patient-1	Male	70	DM, HT, HL, BPH	+
	Patient-2	Male	46	DM, HT	+
	Patient-3	Female	72	DM, HT	−
	Patient-4	Male	62	DM, HT	+
	Patient-5	Male	61	DM, HT, RENAL FAILURE	−
	Patient-6	Male	56	DM, HT, HL	+

**Table 2 jcm-12-01019-t002:** The contraction response of phenylephrine-stimulated venous rings exposed to cumulative sulodexide concentrations only, and in the L-NAME-incubated rings in endothelium-intact and denuded samples. * *p* < 0.05 within-group comparison ** *p* < 0.05 between-groups comparison.

**Phenylephrine Stimulated Only**								
Groups	Phenylephrine	Sulodexide						
	6 × 10^−7^	0.001 mg/mL	0.005 mg/mL	0.01 mg/mL	0.05 mg/mL	0.1 mg/mL	0.5 mg/mL	1.0 mg/mL
Endothel-Intact (*n* = 8)	90.5% ± 0.3	92.6% ± 0.3	94.6% ± 0.3	101.3% ± 0.4	104.0% ± 0.4 *	108.3% ± 0.4 *	110.5% ± 0.4 *	112.9% ± 0.4 *
Endothel-Denuded (*n* = 6)	95.0% ± 0.3	94.9% ± 0.3	94.6% ± 0.3	93.3% ± 0.3	91.9% ± 0.3 **	89.5% ± 0.3 **	87.9% ± 0.3 **	85.2% ± 0.3 **
**L-NAME Pre-incubated (10^−4^ M)**								
Groups	Phenylephrine	Sulodexide						
	6 × 10^−7^	0.001 mg/mL	0.005 mg/mL	0.01 mg/mL	0.05 mg/mL	0.1 mg/mL	0.5 mg/mL	1.0 mg/mL
Endothel-Intact (*n* = 8)	92.3% ± 0.2	91.8% ± 0.3	92.2% ± 0.3	92.5% ± 0.3	92.2% ± 0.3	91.7% ± 0.3	90.2% ± 0.4	89.9% ± 0.3
Endothel-Denuded (*n* = 6)	94.2% ± 0.3	93.6% ± 0.3	93.1% ± 0.2	92.1% ± 0.2	91.0% ± 0.2	89.3% ± 0.2	88.2% ± 0.2	86.8% ± 0.2

## Data Availability

Additional data is unavailable due to ethical restrictions and privacy.

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
