# Peer review of "Sulodexide Develops Contraction in Human Saphenous Vein via Endothelium-Dependent Nitric Oxide Pathway"

_jcm, 2023, doi:10.3390/jcm12031019_

Round 1

Reviewer 1 Report

Manuscript Title: Sulodexide Develops Contraction in Human Saphenous Vein via Endothelium Dependent Nitric Oxide Pathway

Manuscript Number: jcm-2132903

Thank you for this very important research and translational scientific study. I enjoyed the manuscript and the findings.  I have the following questions and recommended changes for the authors.

General comments: You should review the manuscript for proper grammatical structure and clarity. I have listed some examples below that are not clear and the wording contradicts the effect.

a.       Line 50. “Heparan sulfate affects coagulation lesser than unfractionated heparin or low molecular weight heparin, thus decreases the risk of hemorrhage[5]. The sentence is grammatically incorrect by stating coagulation lesser. You could state “to a lesser extent than…”.

b.       Line 54. “It shows anticoagulant activity by inhibiting Factor X and activating thrombin[7].” The sentence indicates that dermatan sulfate in an anticoagulant by inhibiting FX, but then it also activates thrombin which would indicate that it is a procoagulant. Please clarify.

c.       Line 61. “Another fundamental target of SDX is the glycocalyx. Vascular endothelial lumen is coated by it.”. this sentence is not grammatically correct, please revise.

d      There are many other grammatical sentence structures that need correcting. There are many spelling errors that need correcting.

*1. The introduction has a general review about the mechanisms of sulodexide, its clinical effects in arteries, veins, and thrombosis, and references to SAUVIS study. There is also significant review about various inflammatory pathways in chronic venous disease. However, there is a significant paucity of information on contraction, the importance of nitric oxide, and mechanism of venous contraction. The overarching theme of the study is to determine the effects of sulodexide on venous contraction in human saphenous vein. However, there is little to no information with respect to the background that is relevant to the study. The authors should focus briefly on CVD and sulodexide, and bring relevant literature to provide an introduction on venous contraction, venous abnormalities in contraction and implications, and why this is important to study. Clear objectives and a hypothesis would also be important to include.

*2. In the Methods the numerical values and decimal places is inconsistent. It is not possible to accurately have three decimal places for age. The best estimate would be one decimal place.

*3. If the authors could please clarify how many patients provided all of the samples of great saphenous vein? The authors should state the total number of patients, and if all of the same patients provided specimens for each of the experiments. It seems that 8 patients had experiments with intact endothelium and 6 patients had experiments with removal of endothelium. Please clarify in the Methods.

*4. Because many of these patients undergoing coronary revascularization are on anti-hypertensives, statins, antiplatelets, diabetes medications, and antiarrhythmics, how did the authors perform the washout period to ensure that there would be minimal interference (effects) from dwelling medications (e.g., beta-blockers, alpha-blockers, statins)? Were the vein specimens washed multiple times and dwelled in Krebs solution to minimize medication presence and contamination? From time to harvest to time of experiment, what was the time interval? This information should be included in the manuscript.

*5. The authors utilized saline as control vehicle. Why not use Krebs solution?

*6. The authors need to provide information of where sulodexide was obtained and manufacturer.

*7. The authors provide no details of analysis of the data and statistic methodology. This information is essential and needs to be included in the manuscript.

*8. In Figure 1 it is unclear what the numerical values represent. Is this fifty-thousand to one-hundred twenty-thousand percent increases in contraction? If this is 50 to 120 % then the authors cannot have three decimal places, this degree of accuracy is not possible. Please clarify and correct.

*9. The Results section for the explanation of the concentration-dependent contraction to sulodexide is very fragmented and difficult to follow. The authors should provide clarity and state the findings that are concise. The authors should break up the section with headings reflecting the findings for each of the major components of the experiments.

*10. The wording in the figure legends for Figure 1 and Figure 2 is grammatically difficult to follow and incorrect. This need rewording.

*11. In Figure 1 and Figure 3 statistical error bars should be included, with appropriate significance in the figure.

*12. The authors should organize the data for the % contraction and condition with s.d. in tables, and only have narrative of numerical finding in contractions for the central findings. Also, Table 2 and Table 3 s.d. have three decimal places which does not seem possible.

*13. The authors in the Discussion refer to the venotonic effect of sulodexide, but what was actually demonstrated was ex-vivo enhanced contraction. The latter is reflective of the experiments conducted, and the authors should refer to contraction and not venotonic effect which may have additional properties than just contraction. Please correct.

*14. The authors demonstrated that blocking NO synthetase (synthase) with L-NAME, which affects the NO-cGMP pathway, abrogates the contractile effects of sulodexide. The authors did not measure NO via Griess (Nitrite/Nitrate) or with 4-amino-5-methylamino-2′,7′-difluorescein (DAF-FM), an NO-sensitive fluorescent dye. Therefore, the authors should be cautious in making presupposition of the sulodexide effect on NO and venous contraction in the Discussion. The authors should just state the findings that blocking the NO-cGMP pathway with L-NAME and/or denuding the endothelium affects the ability of sulodexide contraction in human veins and may be a result of disruption of NO production or decoupling and impairing the endothelial vasoactive modulation (it is possible that endothelin formation/release is affected and that sulodexide is not able to enhance the synergistic effects of contraction).

*15. Are the authors certain that 0.5 gram tension represents venous hypertension? Usually, saphenous vein tensions can be increased to a base of 2 grams for experimentation. Have the authors actually calculated the conversion to mm Hg in a 0.5 gram tension saphenous vein? In rat IVC (which is much smaller and thinner than saphenous vein) a 0.5 gram tension equated to the pressure generated of 28.4 gram force/cm2 or 20.8 mmHg. The authors should calculate the pressure generated in saphenous vein with a tension force of 0.5 gram (Lim CS, Qiao X, Reslan OM, Xia Y, Raffetto JD, Paleolog E, Davies AH, Khalil RA. Prolonged mechanical stretch is associated with upregulation of hypoxia-inducible factors and reduced contraction in rat inferior vena cava. J Vasc Surg 2011;53:764-73. doi: 10.1016/j.jvs.2010.09.018.).

*16. The authors should provide details of the limitations of this study and not just the samples being form healthy veins.

Reviewer 2 Report

Very interesting article containing a clinically important result. Adding to the well-known antithrombotic and anti-inflammatory properties of sulodexide and its ability to protect glycocalyx, thus preventing the development of endothelial dysfunction, the authors demonstrated the ability of sulodexide to increase the tone of the veins of human lower extremities. Such a set of positive effects makes the drug truly unique.

The article is written clearly and correctly. The only drawback seems to be too long Introduction, in which much space is given to NO-dependent dilatation of the arteries.

In the “Materials and Methods” section it would be useful to indicate the time interval between steps of concentration of sulodexide in the solution was increased. Or the concentration increased simply after steady level achievement? In this case it is interesting how long is the process of the tension increase lasts.

In the “Results” it would be important to indicate how quickly the tension of the venous muscles increased in response to the increases in sulodexide concentration.

Reviewer 3 Report

Well written

Round 2

Reviewer 1 Report

The authors have addressed all of the reviewers questions and concerns, and have made extensive revisions to the manuscript and in the revised format should be considered for publications. 

Thank you